# The spatial and temporal variation of fine particulate matter pollution in Ethiopia: Data from the Atmospheric Composition Analysis Group (1998–2019)

**Ashenafie Bereded Shiferaw** [1]*, **Abera Kumie**[2], **Worku Tefera**[2]

1 Department of Social and Public Health, College of Health Sciences, Debre Tabor University, Debre Tabor, Ethiopia, 2 Department of Environmental and Behavioral Medicine, School of Public Health, College of Health Sciences, Addis Ababa University, Addis Ababa, Ethiopia

* sashenafie@gmail.com

## Abstract

### Background

Evidence suggests ambient fine particulate matter ($PM_{2.5}$) is a risk factor for cardiovascular diseases, lung cancer morbidity and mortality, and all-cause mortality. Countries that implement strong policies are able to reduce ambient $PM_{2.5}$ concentration. In Ethiopia, however, $PM_{2.5}$ monitoring stations, laboratory technicians, and equipment are staggeringly limited. In this study, the spatial and temporal variation of $PM_{2.5}$ in Ethiopia was assessed.

### Methods

Satellite-based $PM_{2.5}$ estimates, from the year 1998 to 2019, by Atmospheric Composition Analysis Group (ACAG) at a spatial resolution of 0.01° X 0.01° was used. The annual mean $PM_{2.5}$ concentration for all administrative regions and zones in Ethiopia was extracted. The average mean from the twenty-two years was also calculated. The trend of $PM_{2.5}$ concentration was graphed and quantitatively calculated using the Mann-Kendall test. The slope of the change over time was estimated using the Theil-Sen slope. At the zonal administration level, for the average annual mean, spatial dependency using univariate Global Moran's I and clustering and outlier tests using Anselin Local Moran's were performed.

### Results

The country's average annual mean $PM_{2.5}$ concentration was 17 $\mu gm^{-3}$. The Afar region had the highest concentration, 27.9 $\mu gm^{-3}$. The Mann-Kendall S was positive and significant at $p < 0.001$. The spatial distribution of satellite-based ambient $PM_{2.5}$ concentration was non-random. Significant highest value clustering of ambient total $PM_{2.5}$ concentration exists in the Afar, Eastern Tigray, and Eastern and Southeastern Amhara while the significant lowest value dispersing was observed in the Southern Oromia and Somali region.

**Data Availability Statement:** All relevant data are within the paper and its Supporting Information files. Furthermore, the datasets used can be

accessed and downloaded from https://sites.wustl.edu/acag/datasets/surface-pm2-5/.

**Funding:** The authors received no specific funding for this work.

**Abbreviations:** ACAG, Atmospheric Composition Analysis Group; EPA, Environmental Protection Agency; EU, European Union; GEOS, Chem Goddard Earth Observing System-Chemical Transport Model; LMICs, low- and middle-income countries; LRTI, Lower Respiratory Tract Infection; MISR, Multi-angle Imaging SpectroRadiometer; MODIS, Moderate Resolution Imaging Spectroradiometer; NASA, National Aeronautics and Space Administration; PBL, Planetary Boundary Layer; PM, Particulate Matter; PM$_{2.5}$, Particulate Matter with mass concentration size below 2.5 μm diameter; RH, Relative Humidity; SeaWIFS, Sea-viewing Wide Field-of-view Sensor; SNNPR, Southern Nations Nationalities and Peoples Region; URTI, Upper Respiratory Tract Infection; US, United States; WHO, World Health Organization.

## Conclusion

At the national and regional levels, the annual mean ambient PM$_{2.5}$ concentration is beyond the World Health Organization (WHO)-recommended level. The ambient PM$_{2.5}$ concentration distribution is spatially dependent and significantly clustered in space. Installation of additional ground-based PM$_{2.5}$ monitoring devices, particularly in regions where PM$_{2.5}$ concentration is higher, is recommended. Validating satellite-based PM$_{2.5}$ data with ground-based measurements in the country is also advised.

## Introduction

Particulate matter (PM) is a composition of chemical compounds and biological agents. It is a collection of solids and liquids hung in the indoor or outdoor air. Through various human activities (i.e. engine combustion in vehicles, burning biomass and other solid fuels, agricultural and industrial activities, and human-induced dust release) or natural causes (i.e. blown dust from arid areas) PM can be released straight into the air or formed from emitted pollutants. Chemicals constituted in PM include organic carbon, elemental carbon, soil dust, sulfate, nitrate, ammonium ions, and heavy metals (including lead, cobalt, copper, arsenic, zinc, manganese, nickel, aluminum, thallium, antimony, vanadium, chromium and cadmium). The biological constituents (bioaerosols) of PM include allergens and microbial compounds of bacterial and fungal origins [1–8].

Fine particulate matter (PM$_{2.5}$), mass concentration size below 2.5 μm diameter, can pass the blood gas barrier to circulate in the blood system and reach beyond pulmonary organs such as the liver and kidney. Chemicals constituted in outdoor PM$_{2.5}$ will have a chance to accumulate, damage endothelial cells, and cause systemic inflammation [9]. Fine particulate matter is known to be associated with childhood pneumonia [10, 11], upper respiratory tract infection (URTI) [10], lower respiratory tract infection (LRTI) [11, 12], and increased viral and bacterial (mycoplasma pneumonia) among children [13]. Evidence also shows it is associated with lung cancer morbidity and mortality, cardiovascular-related mortality [14–16], and all-cause mortality [17–20].

The concentration of outdoor PM$_{2.5}$ is influenced by the intensity of rainfall, the duration of rainfall, and the relative humidity of the environment. Evidence shows that the scavenging role of rainfall to remove PM$_{2.5}$ is greater during heavy rainfall and lower humidity. In areas where the air quality is poor and the initial concentration of PM$_{2.5}$ is higher the scavenging role of rainfall increases. In light rainfall and higher humidity, the concentration of PM$_{2.5}$ increases [21–23]. Source emission increases on rainy and snowy days and when coupled with reductions in the atmospheric vertical convection the level of PM$_{2.5}$ also increases [24]. The dispersion and dilution of PM$_{2.5}$ determine its concentration. Both are affected by the height of the Planetary Boundary Layer (PBL). The lowest PBL height is consistent with the higher and maximum concentration of PM$_{2.5}$. The stable inversion above PBL as a result of warm and humid airflow, long-wave radiation cooling during the night, solar-radiation reduction during the daytime, or aerosol radiative effect during heavy air pollution constrain the height of PBL and leads to an increase in PM$_{2.5}$ concentration [25–28]. An increase in outdoor PM$_{2.5}$ levels was recorded during the winter—in Japan [3] and China [27], rainy—in Addis Ababa [4], and spring seasons (in China) [29] than in summer and dry seasons. The increased use of fuel for heating during the colder season explains it [4, 24] and that winter's stable atmosphere, cold, and meteorology promote a longer life for PM$_{2.5}$ in the atmosphere [3, 27].

The concentration of $PM_{2.5}$ also differs from country to country and from one region of the world to another. Low- and middle-income countries (LMICs) are suffering from poor outdoor air quality. In the past decades, for instance, the ambient $PM_{2.5}$ concentration has kept increasing in the city of Kathmandu, Nepal [30] and Dhaka, Bangladesh [31]. Due to the degree of pollution, the majority of days in these cities are reported to be unhealthy for humans [30, 31]. In 2017, southeast Asia countries (Nepal, India, Bangladesh, Pakistan, and China) and western sub-Saharan African countries (Niger, Cameroon, Nigeria, Chad, and Mauritania) showed the highest concentrations of $PM_{2.5}$. Almost no population in China, India, Pakistan, and Bangladesh breathes air below the WHO's (World Health Organization) annual exposure limit of $PM_{2.5}-10$ $\mu gm^{-3}$ [32, 33]. The situation in Africa is not different from China and India in terms of breathing polluted air. According to WHO, in 2016, all under-five children in Africa and Eastern Mediterranean were exposed to ambient $PM_{2.5}$ greater than 10 $\mu gm^{-3}$ [34].

The sources of $PM_{2.5}$ pollution also vary across regions. Windblown dust from the Sahara Desert in Western Africa, Northern Africa, and the Middle East is one of the major contributors. Dust from construction, burning of coal in industry and power plants, transportation, and burning of solid fuels for households are the major source of $PM_{2.5}$ in India and China. Improvements in air quality were seen in countries where strong policies are implemented to reduce air pollution. United States (US), Brazil, European Union (EU), Japan, Russia, Mexico, and Indonesia have made a relatively large decline in ambient $PM_{2.5}$ concentration between 1990 to 2017 [32, 33].

However, the problem persists in most LMICs where air quality monitoring systems are available inadequately as they incur large investments and running costs that these countries usually may not afford [35]. In 2021, WHO updated the 2005 air quality guideline and suggested a 15 $\mu gm^{-3}$ and 5 $\mu gm^{-3}$ ambient $PM_{2.5}$ exposure limit for 24 hours and annual mean, respectively [36]. In Ethiopia, the government passed policies (such as Environmental pollution control proclamation No.300/2002) that forbid the release of contaminants into the environment, including the air. These policies also state the government must regulate, monitor, and oversee compliance. However, the government lacks enough laboratories and skilled professionals to monitor air quality. As a result, evidence that influences meaningful policy actions and record-keeping regarding air pollution is undersupplied [37]. To the best of our knowledge, no previous study at the national level in the country characterizes the geographical and temporal distribution of ambient $PM_{2.5}$. This study, therefore, has assessed the spatial distribution and temporal variation of the concentration of ambient $PM_{2.5}$ in Ethiopia.

## Methods

### Study area and study design

The study area is the country Ethiopia which is found in eastern Africa between 3˚24' to 14˚15' North and 33˚00' to 48˚00' East. Its population is more than one hundred ten million with 1.1 million square kilometers of land area. The country is divided into two federal city administrations and nine regional states [38]. The second administrative unit next to regional states is Zone which in turn has Woredas as a third-level administrative unit. The lowest administrative unit next to Woreda in the country is Kebele [39].

### Variables

The dependent variable is the total $PM_{2.5}$ concentration measured in micrograms (µg) per cubic meter ($m^{-3}$) of atmospheric air. This study used the annual mean concentration of ambient $PM_{2.5}$. The independent variables were space and time in years.

## Data source and measurement

Areal $PM_{2.5}$ can be measured by devices installed at sampling/monitoring stations on land in a municipal city or state [29, 40–42]. The devices should be protected from rain and unfavorable weather and can be placed on a rooftop [42] or two meters above the ground [4]. The gravimetric method technique determines PM concentration by subtracting the filter weight before sampling from the weight after sampling [43]. While the real-time monitor—Beta-Attenuation Monitor (BAM) measures mass particle concentration by taking the before and after difference in the attenuation of the beta radiation that is emitted and passes through the rolling filter tape [42, 44–46]. The BAM device is approved by the United States (US) Environmental Protection Agency (EPA) [47]. The detailed mounting procedure is published elsewhere [46, 48]. Nowadays, remote sensing devices such as satellites are being used in places where ground-based monitoring is absent [49].

Due to the absence of ground-based $PM_{2.5}$ monitoring stations across the country, this study used satellite-based $PM_{2.5}$ estimates produced by the Atmospheric Composition Analysis Group (ACAG) at Dalhousie University and Washington University [50]. This study used the $PM_{2.5}$ concentration estimate by the ACAG at a spatial resolution of $0.01° \times 0.01°$. The group retrieved aerosol data collected using multiple instruments—MODIS, MISR, and SeaWIFS—mounted on National Aeronautics and Space Administration (NASA's) satellites. Based on the chemical transport model (from GEOS-Chem) the geophysical relationship between Aerosol Optical Depth (AOD) and surface $PM_{2.5}$ was declared. The estimate of surface geophysical $PM_{2.5}$ was determined by the retrieved total column AOD accuracy and the simulated relationship between AOD and $PM_{2.5}$ represented by $\eta$ (eta). Surface $PM_{2.5}$ was computed as the product of retrieved total column AOD and $\eta$. In calculating $\eta$, variables of aerosol vertical structure and optical properties, hygroscopicity, emission, composition, and factors of sampling time and meteorological characteristics at the local level (wind speed and direction, boundary layer height, and relative humidity (RH)) were included [50]. Daily collected values were used to calculate the monthly mean AOD values. Daily values that are outliers—above the sum of the local mean and its two standard deviations (SD) within $111^{*}111$ kilometers—and without 25% or above local coverage were removed. Missing daily values were approximated using the average of available collected values on the missing days and accounting for the month's assigned climatologic ratio. Local coverage of less than 50% monthly mean was removed and interpolation was made from the same month of other years. The monthly mean $PM_{2.5}$ was derived from the monthly mean AOD. The annual mean $PM_{2.5}$ was then computed from the monthly mean concentrations. The ACAG also validate their estimate with land-based $PM_{2.5}$ monitoring systems by employing Geographically Weighted Regression (GWR). Further information about the techniques used is published elsewhere [50, 51].

The satellite-based $PM_{2.5}$ for this study was downloaded from the ACAG website, https://sites.wustl.edu/acag/datasets/surface-pm2-5/. It was available for the public for free and requires no registration. The annual mean $PM_{2.5}$ concentration, from the year 1998 to 2019, was extracted from the downloaded dataset of version V5.GL.02 for all administrative regions and zones in the study area.

## Data management procedure

The software ArcGIS v 10.7 was used to view the downloaded spatial dataset, clip raster, and perform zonal statistics. The software GeoDa v 1.20.0.8 on the other hand was used to perform the spatial dependency and clustering tests. Microsoft Excel 2016 software was also used to graph the trend of annual mean ambient $PM_{2.5}$ concentration across regions.

## Data analysis procedure

The average mean was calculated from the twenty-two years' annual mean $PM_{2.5}$ concentration for the 11 regions and 79 zones. The trend of annual mean $PM_{2.5}$ concentration across regions was graphed in Microsoft Excel. The significance and direction of change in ambient $PM_{2.5}$ concentration over time were quantitatively calculated using the Mann-Kendall monotonic trend analysis test. The slope of the time-series change was estimated using the Theil-Sen slope. Mann-Kendall is commonly used to analyze environmental data such as trends in climate change [52].

At the zonal administration level, for the average annual mean, spatial dependency and clustering tests were performed. Using univariate Global Moran's I statistic measure the distribution of mean annual $PM_{2.5}$ was determined whether the distribution was dispersed, clustered, or random. Univariate Global Moran's I is a technique that determines the spatial autocorrelation in the values of a specific variable (ambient $PM_{2.5}$ concentration) between different geographic features using location (neighbor) and the values of the variable in each feature as inputs [53]. Then, using the Anselin Local Moran's I clustering and outlier analysis, features univariate value similarity and dissimilarity with their neighbor features, were determined. The similarity between a feature univariate value and its neighboring features shows clustering (high or low values clustering). Dissimilarity of a feature from its neighboring features on the other hand shows the feature is an outlier. A Z-score of a high positive value indicates a statistically significant clustering while a low negative value indicates a statistically significant spatial outlier [54].

## Ethical statement

The global estimate of annual mean $PM_{2.5}$ concentration was made accessible to the public by the ACAG and used for free. Furthermore, this study was approved by the College of Health Sciences of Addis Ababa University Ethical Review Committee with a reference number SPH/1119/13.

## Results

This study has described the annual mean ambient $PM_{2.5}$ concentration of the 11 regions and 79 zones in Ethiopia from the year 1998 to 2019. Over the years, the average annual mean of total $PM_{2.5}$ concentration in the country was 17 $\mu gm^{-3}$ and ranges from 13.9 to 20.2 $\mu gm^{-3}$. Afar region had the highest average annual mean $PM_{2.5}$ concentration (from 1998–2019), 27.9 $\mu gm^{-3}$ followed by Tigray (21.4 $\mu gm^{-3}$), Amhara (21.2 $\mu gm^{-3}$), Gambela (20.9$\mu gm^{-3}$), Benishangul-Gumuz (20.4 $\mu gm^{-3}$), and Addis Ababa (20.3 $\mu gm^{-3}$) (Fig 1).

## The trend in $PM_{2.5}$ concentration across different regions

Based on the analysis of the presented data, the Mann-Kendall S was positive and significant at p<0.001, indicating that in Ethiopia, the annual mean $PM_{2.5}$ has been on an increasing trend over the years (1998 to 2019). The Theil-Sen slope was found to be 0.21. Afar, Amhara, Tigray, Gambela, Benishangul-Gumuz, and Addis Ababa regions showed a consistently higher $PM_{2.5}$ concentration compared to other regions. Afar and Somali regions had the highest and lowest $PM_{2.5}$ concentration across the years, respectively. In 2014, across all regions lower $PM_{2.5}$ concentration was observed (Fig 2).

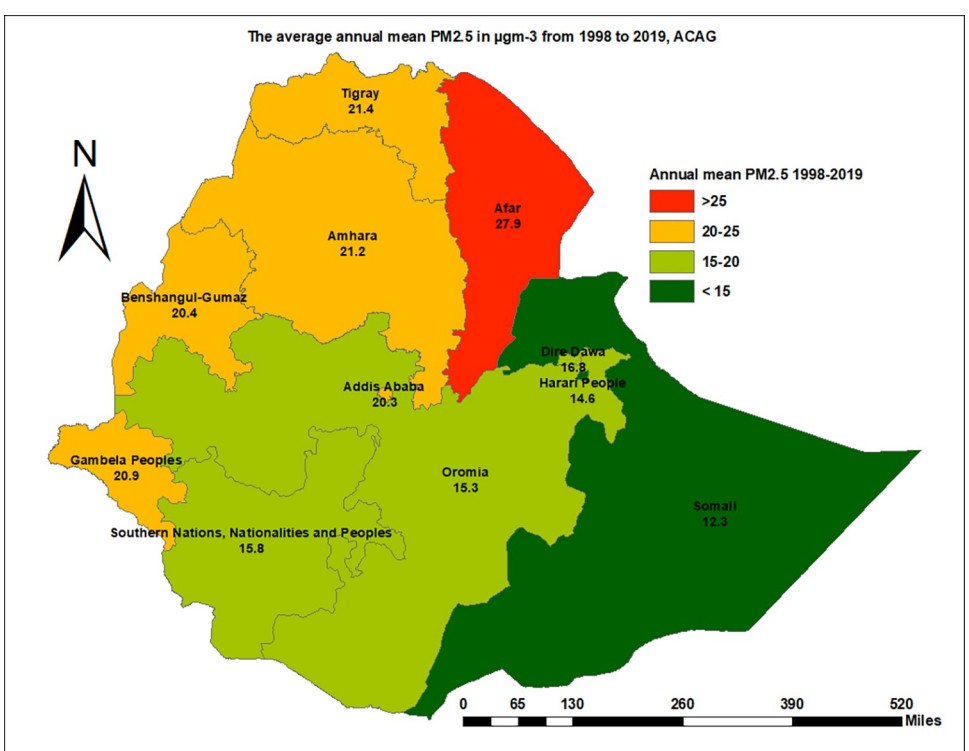

**Fig 1. Ethiopia regions average annual mean fine PM concentration in μgm$^{-3}$ from 1998–2019, ACAG (Source: Shapefile from openAFRICA 2013: https://gadm.org/).**

## Variability of PM$_{2.5}$ concentration within each region

In the past decades, the variability of PM$_{2.5}$ concentration within a region was highest in Afar, Addis Ababa, Amhara, Dire Dawa, Harari, and Benishangul-Gumuz regions, while the lowest variability is in Somali, Tigray, Gambela, Southern Nations Nationalities and Peoples, and Oromia (Fig 3).

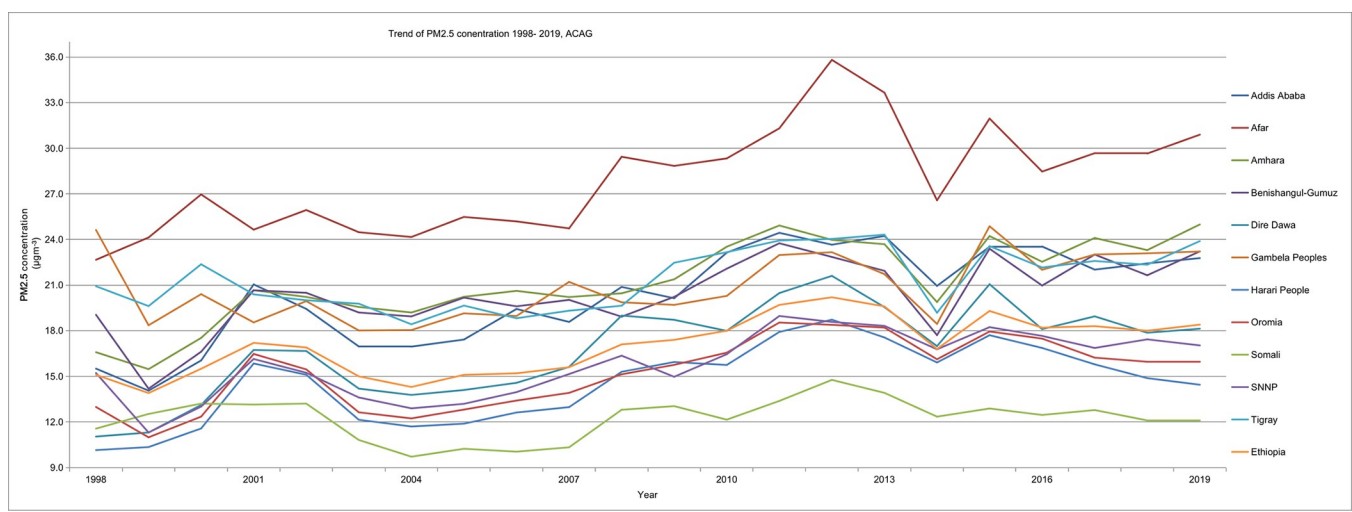

**Fig 2. Trends of fine PM concentration in μgm$^{-3}$ across regions in Ethiopia, ACAG 1998–2019.**

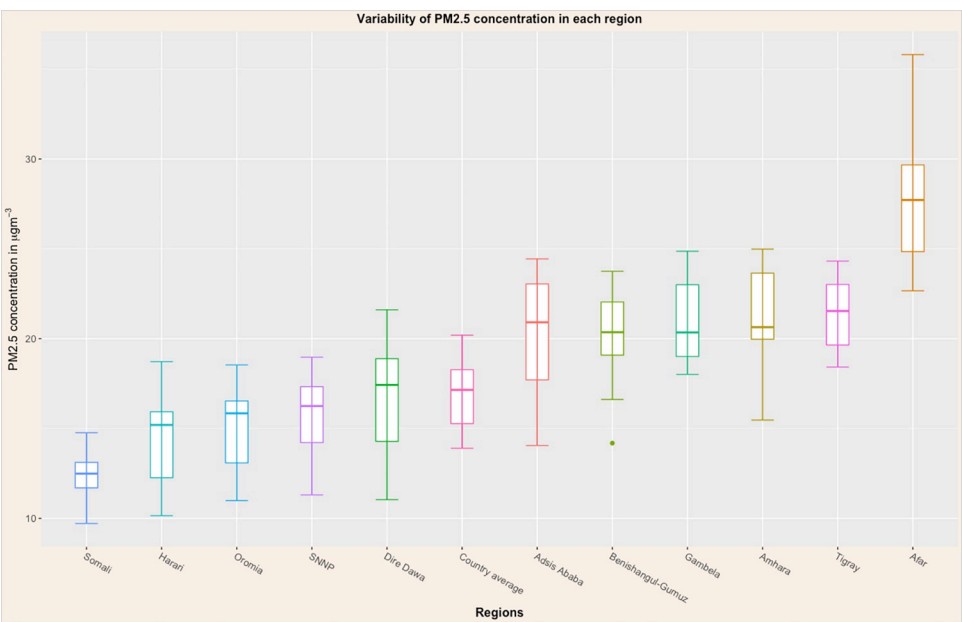

**Fig 3. Variability of fine PM concentration in µgm-3 within each region in Ethiopia, ACAG 1998–2019.**

## Spatial distribution of $PM_{2.5}$ concentration

Across the geographical areas of Ethiopia, the annual mean $PM_{2.5}$ concentration is distributed non-random. This indicates that overall the concentration of $PM_{2.5}$ follows a complete spatial pattern. The twenty-two years (1998–2019) average annual mean $PM_{2.5}$ concentration univariate global Moran's I value was 0.86 ($p$-value = 0.001). The result demonstrates the geographical relationships among Zones (second administrative units) determine the variation in the concentration of $PM_{2.5}$.

## Spatial significance and clustering and outlier analysis of $PM_{2.5}$ concentration

Significant spatial dependency in $PM_{2.5}$ concentration was observed in Afar, Central and Eastern Tigray, Eastern and Northern Amhara, Somali, Eastern and Southern Oromia, and Southern SNNP regions. These are the specific Zones where the concentration of $PM_{2.5}$ is dependent on the spatial relationships among Zones ($p$-value < 0.05). In the other Zones, there was a non-significant influence of geographical relationships on the $PM_{2.5}$ concentration (Fig 4).

The higher or clustered $PM_{2.5}$ concentration was in Afar, Central and Eastern Tigray, and Eastern and Northern Amhara ($p$-value < 0.05). In the clustered type of spatial association, Zones with higher ambient $PM_{2.5}$ concentration attract similar Zones in their neighborhood. In contrast, the lowest or dispersed $PM_{2.5}$ concentration was in Eastern and Southern Oromia, and Southern SNNPR ($p$-value < 0.05). In the dispersed type of spatial association, the neighborhood of Zones with lower ambient $PM_{2.5}$ concentration is surrounded by Zones with lower ambient $PM_{2.5}$ concentration (Fig 5). In this study, there were no observed outlier Zones (High-Low or Low-High). The High-Low Outlier Zone is with a higher $PM_{2.5}$ concentration surrounded by neighbor Zones with a lower $PM_{2.5}$ concentration. The Low-High Zone, on the other hand, has a lower $PM_{2.5}$ concentration and is surrounded by neighboring Zones with higher $PM_{2.5}$ concentration.

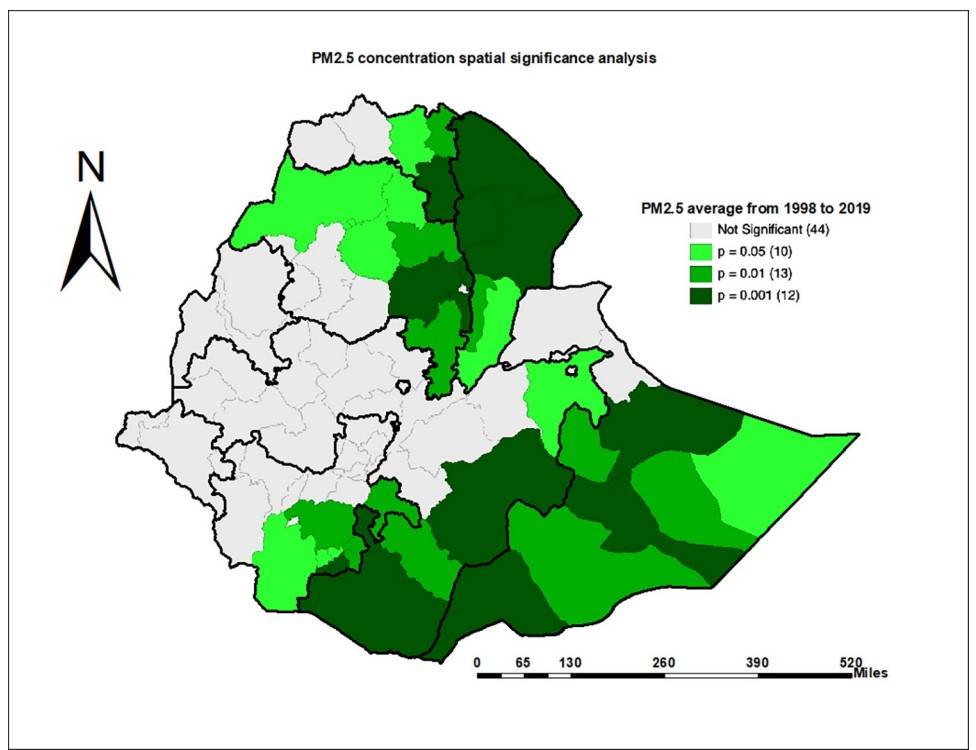

**Fig 4. Average annual fine PM concentration spatial dependency significance test, ACAG 1998–2019 (Source: Shapefile from openAFRICA 2013: https://gadm.org/).**

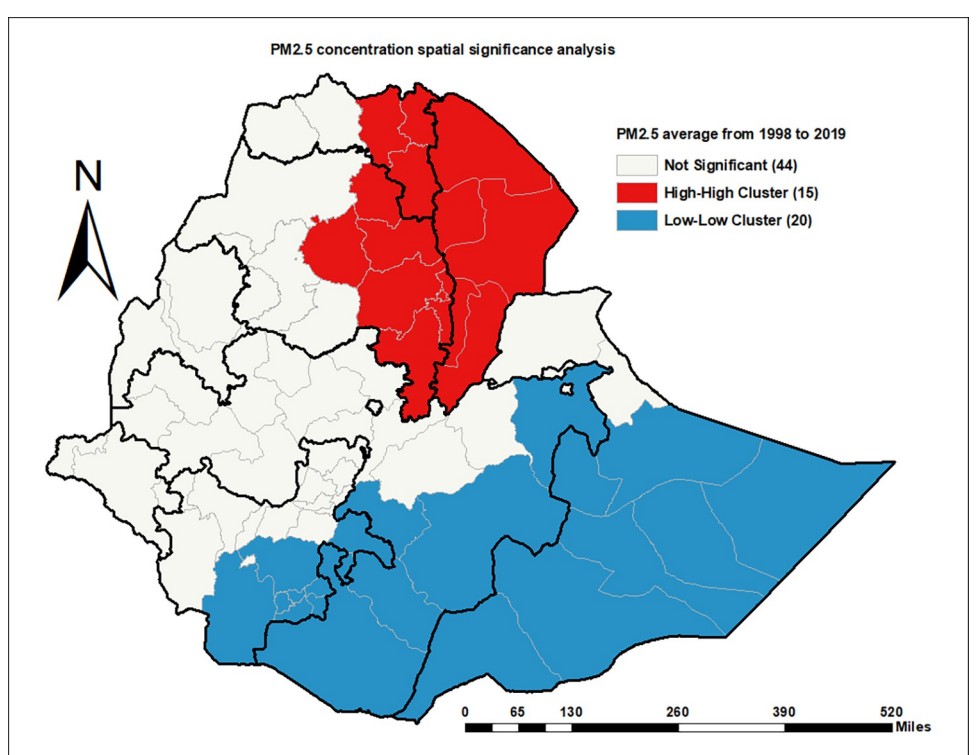

**Fig 5. Average annual fine PM concentration spatial clustering and outlier analysis, ACAG 1998–2019 (Source: Shapefile from openAFRICA 2013: https://gadm.org/).**

## Discussion

Overall, all regions, in all years, had an annual mean ambient $PM_{2.5}$ concentration higher than the WHO's interim air quality guideline recommendation (not exceeding an annual average concentration of 10 $\mu gm^{-3}$). The country's average annual mean $PM_{2.5}$ concentration was 17 $\mu gm^{-3}$. There was a significant increasing trend in $PM_{2.5}$ concentration. The Afar region had the highest concentration, 27.9 $\mu gm^{-3}$. The spatial distribution of satellite-based ambient $PM_{2.5}$ concentration was non-random.

This study's result, a consistent above 10$\mu gm^{-3}$—the WHO recommendation for mean annual ambient $PM_{2.5}$ concentration—across all regions, was coherent with previous studies in developing countries like China [15] and Africa [34, 55]. Higher levels of ambient $PM_{2.5}$ is expected with fast population growth [33, 56], the use of solid fuel for cooking [33], and limited air quality monitoring and control measures [35]. However, in Ethiopia, ambient $PM_{2.5}$ concentration is showing no reduction while countries like China, the EU, the US, Indonesia, Mexico, and Japan were able to enforce policies that resulted in a significant decline in the concentration of ambient $PM_{2.5}$ [32, 33]. Most developing countries, including Ethiopia, tend to have limited air quality monitoring systems [35, 57]. There is no strong system in the country that keeps track of ambient $PM_{2.5}$ and the consequences associated with it [57]. There is evidence that says the level of pollution is increasing in Ethiopia with increasing urbanization, industrialization, and traffic [32].

The consistently higher $PM_{2.5}$ concentration in the regions of Amhara, Tigray, Addis Ababa, Gambela, and Benishangul-Gumuz than in other regions might be a result of a combination of different factors. Amhara and Tigray are among the regions with higher population density [58], lower wind speed [59], and terrain described as mountainous with gorges, valleys, and rugged formations [60]. Empirical evidence shows areas with higher population density experienced higher levels of pollution [61, 62]. The population density is associated with increased energy consumption which in turn is the major source of $PM_{2.5}$ pollution [63]. Local meteorology (PBL height, temperature, RH, and wind speed and direction) and topography influence the concentration and dispersion of $PM_{2.5}$ [64]. A study found that a small difference in wind speed (as small as one to two meters per second) between areas is associated with a reduced concentration of pollution [65]. Wind speed might be one of the reasons for the higher $PM_{2.5}$ in the Gambela, Benishangul-Gumuz, Amhara, and Tigray regions considering the lower wind speed in these regions relative to the other areas. Topography such as deep basins and valleys are usually associated with elevated $PM_{2.5}$ concentration by creating unique conditions of meteorology and atmospheric vertical structure [66]. Such terrains are common in the Amhara and Tigray regions while flat terrains are prominent in Oromia and Somalia regions [60]. Compared to the other regions, in the past two decades, Gambela and Benishangul-Gumuz regions experienced a large loss of forests relative to their geographical size to increasing farming lands [67]. Both destruction of forests and investment in agriculture are known contributors to increased pollution [68].

In this study, over the twenty-two years, within a region fluctuation in the annual mean $PM_{2.5}$ concentration was observed in Afar, Addis Ababa, Amhara, Dire Dawa, Harari, and Benishangul-Gumuz regions. The authors recommend further investigations into the specific events and factors which might influence the within-region variability of ambient $PM_{2.5}$ concentration.

The high clustering of satellite-based ambient $PM_{2.5}$ in the Eastern and Southeastern Amhara, Eastern Tigray, and Afar areas might be attributed to the country's major import-export highway route passing through these areas [69]. A one-year study with weekly $PM_{2.5}$ sampling (n = 62) in Addis Ababa indicated that traffic flows contributed to 28% of the mass

of $PM_{2.5}$, which perhaps might be aligned with the sources of import-export highway vehicles [70]. Additionally, due to $PM_{2.5}$'s nature as a regional pollutant, it is transported to other nearby areas [33]. The transport sector is one of the major contributors to the increasing ambient $PM_{2.5}$ [71]. In Afar and Eastern Tigray, in particular, the consumption of agricultural residues (animal dung and crop stalks and stubbles) as cooking fuel is higher due to the scarcity of woody biomass [72]. Biomass sources are the second biggest contributors to $PM_{2.5}$ in cities of developing countries [73] including Addis Ababa (18.3%) [70]. The Afar region is a relatively arid area with less tree coverage and is home to the Danakil desert. Desert areas like the Danakil desert, semi-arid, and arid areas are known to be hit by sand and dust storms with the potential to reach neighboring regions or even countries [74]. Dust and sands are the major sources as well as constituents of $PM_{2.5}$ [32, 33, 70]. However, the authors still believe a further investigation into the country-specific sources and drivers of $PM_{2.5}$ is needed.

### Implication of the study

The decadal pattern of $PM_{2.5}$ concentration has been increasing steadily, and its spatial distribution is unequal across the country. There were no outlier zones, indicating the $PM_{2.5}$ concentration in neighboring Zones is relatively similar. The importance of taking into account these key results when developing meaningful strategies for reducing $PM_{2.5}$ concentrations and monitoring cannot be overstated.

### Strengths and limitations

To the best of our knowledge, this study portrayed the spatial and temporal distribution of ambient $PM_{2.5}$ concentration at the national level for the first time in the country. This study is strong in using the available ambient $PM_{2.5}$ data to further analyze and provide more information. As a result, the findings may have implications for the national and other stakeholders' policies regarding air quality improvement. The results also offer insights that the installation of a ground-based monitoring system to consider the spatial dependency of ambient $PM_{2.5}$ concentration.

The study is limited in not accounting for measurement errors attributed to satellite-based $PM_{2.5}$ estimates. Due to the absence of ground-based monitors for adjustment and validation in Ethiopia measurement bias is introduced while estimating $PM_{2.5}$ from satellite-retrieved data. In areas or regions where the concentration of $PM_{2.5}$ is lower the higher the root-mean-square error. There is a consistent relationship between ground-based monitored and validated $PM_{2.5}$ across all regions. However, regions with fewer ground monitors (Africa, Latin America, and Asia) showed the largest uncertainties [50]. Compared to areas with dense monitors in under-monitored areas (India, Africa, Middle-East, and South America), the agreement between validated $PM_{2.5}$ and ground-based monitored increases when the density of monitors increases [75].

### Conclusions

Both at the national and regional levels, the annual mean ambient $PM_{2.5}$ concentration is $>10\mu gm^{-3}$—above the WHO interim guideline. This study identified a significant spatial disparity and clustering of the distribution of ambient $PM_{2.5}$ concentration. The authors recommend the installation of ground-based monitoring systems across the country. Improvement in the rigorous satellite-based $PM_{2.5}$ estimation using the existing ground-based measurements in Ethiopia is invaluable. It is also important to examine sources of ambient $PM_{2.5}$ and the magnitude of their contribution in areas where $PM_{2.5}$ concentration is clustered.

## Supporting information

**S1 File.**
(ZIP)

**S2 File.**
(ZIP)

**S3 File.**
(ZIP)

**S1 Table. Regional level annual mean PM$_{2.5}$ in μgm$^{-3}$ in table.**
(DOCX)

## Acknowledgments

The authors are thankful to Samson Wakuma for the polite responses to facilitate the administrative issues.

## Author Contributions

**Conceptualization:** Ashenafie Bereded Shiferaw.

**Data curation:** Ashenafie Bereded Shiferaw.

**Formal analysis:** Ashenafie Bereded Shiferaw.

**Investigation:** Ashenafie Bereded Shiferaw.

**Methodology:** Ashenafie Bereded Shiferaw.

**Resources:** Ashenafie Bereded Shiferaw.

**Software:** Ashenafie Bereded Shiferaw.

**Supervision:** Abera Kumie, Worku Tefera.

**Validation:** Ashenafie Bereded Shiferaw, Abera Kumie, Worku Tefera.

**Visualization:** Ashenafie Bereded Shiferaw.

**Writing – original draft:** Ashenafie Bereded Shiferaw.

**Writing – review & editing:** Ashenafie Bereded Shiferaw, Abera Kumie, Worku Tefera.

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
