## [Decision Letter · Decision Letter 0]

27 Dec 2022

PONE-D-22-31123The spatial and temporal variation of fine particulate matter pollution in Ethiopia: Analysis of the Atmospheric Composition Analysis Group 1998 to 2019PLOS ONE

Dear Dr. Shiferaw,

Thank you for submitting your manuscript to PLOS ONE. After careful consideration, we feel that it has merit but does not fully meet PLOS ONE’s publication criteria as it currently stands. Therefore, we invite you to submit a revised version of the manuscript that addresses the points raised during the review process.

We look forward to receiving your revised manuscript.

Kind regards,

Basant Giri, Ph.D.

Academic Editor

PLOS ONE

Journal Requirements:

Reviewers' comments:

Reviewer's Responses to Questions

**Comments to the Author**

1. Is the manuscript technically sound, and do the data support the conclusions?

Reviewer #1: Yes

Reviewer #2: Yes

2. Has the statistical analysis been performed appropriately and rigorously? 

Reviewer #1: Yes

Reviewer #2: Yes

3. Have the authors made all data underlying the findings in their manuscript fully available?

Reviewer #1: Yes

Reviewer #2: Yes

4. Is the manuscript presented in an intelligible fashion and written in standard English?

Reviewer #1: Yes

Reviewer #2: No

5. Review Comments to the Author

Reviewer #1: This study assesses the spatial and temporal variation of PM2.5 in Ethiopia based on satellite derived estimates of PM2.5 by the Atmospheric Composition Analysis Group. Annual mean PM2.5 concentration from 1998 to 2019 for all administrative regions and zones in Ethiopia is used. Since ground-based measurements of PM2.5 are limited, satellite-based estimates can provide important information on the geographical distribution of particulate matter pollution in both time and space.

Although the analysis presented is sound, the description is not detailed enough. There needs to be a more thorough description of the results and their implications as well as a detailed background and context in the results section. Therefore, I recommend major revision before this manuscript can be accepted for publication in PLOS ONE.

General comments:

1. Results section: This section is too limited. It needs more details to add context to the results. For example, what does "significant spatial dependency" mean? What does it mean to say the PM concentrations are distributed non-random, clustered or dispersed. These results should be quantified and explained in more detail.

2. Strengths and limitations section: Even though the satellite derived PM2.5 product has not been validated in the study region due to the lack of ground observations, it must have undergone validation by measurements in other locations. What are the conclusions of those validations. The authors need to discuss the general uncertainty in this product using literature review.

3. The manuscript will improve with proof-reading and correcting grammatical and language errors.

Specific comments:

Line 28: This sentence is missing a verb. You might add "are" after "strong policies".

Line 30: It would be better to add "in Ethiopia" after "variation of PM2.5" in this sentence.

Line 34: It is not clear why the “six years” is mentioned here. This is not consistent with the text which shows results for 22 years.

Line 35: “Mann-Kendal” should be “Mann-Kendall”. Correct this spelling elsewhere too.

Line 44: Consider changing "has existed" to "exists".

Line 48: Consider changing 'health-recommended' to 'WHO-recommended'.

Line 49: Sentence fragment. Rewrite.

Line 94: “Outdoor PM2.5 level increases during winter, rainy, and spring seasons (3,4,18) than in summer and dry seasons”

Is this a general statement or is it true for specific locations? Rainy seasons should see lower PM levels due to the scavenging of particulates from rain drops and cloud droplets. This should be clarified.

Line 99 – Line 102: This statement requires citations. Here is a recent example with a focus on a South Asian country: https://doi.org/10.1016/j.atmosres.2021.105623

Line 110: Subscript PM2.5 to be consistent with other places where it appears.

Line 166- Line 168: This statement should cite the ACAG satellite-based PM2.5 product publication.

Line 268: Define what the high-high, low-low, high-low and low-high cluster and outliers mean.

Reviewer #2: PLOS ONE

The spatial and temporal variation of fine particulate matter pollution in Ethiopia:

Analysis of the Atmospheric Composition Analysis Group 1998 to 2019

Manuscript Number: PONE-D-22-31123

General Comment

This paper is related to work, there is evidence that ambient fine particulate matter (PM2.5) raises the risk of heart disease, lung cancer, and all-cause death. Strongly regulated nations can lower the concentration of ambient PM2.5. However, Ethiopia has shockingly few PM2.5 monitoring stations, laboratory staff, and resources. The regional and temporal fluctuation of PM2.5 was evaluated in this study. The annual mean ambient PM2.5 concentration is higher than the amount recommended for health at the national and regional levels. The distribution of ambient PM2.5 concentrations is significantly geographically clustered and spatially dependent. Additional ground-based PM2.5 monitoring equipment, especially in areas with higher PM2.5 concentrations. Additionally, it is recommended to compare ground-based observations in the nation with satellite-based PM data. This work is good and related to my field. All the figures and tables consist to this manuscript is looking nice. I enjoy to read this manuscript, however the addition of few citations and grammatical, rephrase and several my below (specific comment) definitely improve this work. In my opinion, there are some minor suggestions /recommendations that will help to improve the quality of manuscript before acceptance.

Specific comments:

(1) The collection of lots number of Keywords, authors are requested to delete few and keep the genesis of this work.

(2) The numbering in the introduction should be avoided in the following lines:

PM can be released straight into the air or formed from emitted pollutants (1). Chemicals constituted in PM include organic carbon, elemental carbon, soil dust, sulfate, nitrate, ammonium ions (2–4), ozone (5), and heavy metals (i.e. chromium and cadmium) (6). The biological constituents (bioaerosols) of PM2.5 include allergens and microbial compounds of bacterial and fungal origins (1,7).

(3) Please change/modify the references in the following:

“all-cause mortality (11,12,14–17)”

(4) Rephrase the following sentences

It obligates the government to regulate, monitor, and oversee compliance. However, the government lacks enough monitoring laboratories and skilled professionals to monitor air quality. As a result, evidence that influences meaningful policy actions and record-keeping regarding air pollution is undersupplied (24). To the best of our knowledge, no study in the country characterizes the geographical distribution of ambient PM2.5. This study, therefore, has assessed the spatial and temporal variation of PM2.5 in Ethiopia.

(5) Abbreviations may be shifted into the starting of the manuscript.

(6) There are lots of commas and full stops are missing.

(7) Improve fig 1

(8) The current study may be reflect strong impact if you can cite the following reference.

-Characterization of PM10 over urban and rural sites of Rajnandgaon, central India, Natural Hazards

- Review on composition, emission sources of RSPM, TSPM, heavy metals and ions with effect on environment and health, Res. J. Chem. Environ

-Understanding Sources and Composition of Black Carbon and PM2.5 in Urban Environments in East India

- Toxicity and health risk assessment of polycyclic aromatic hydrocarbons in surface water, sediments and groundwater vulnerability in Damodar River Basin, Groundwater for Sustainable Development 13, 100553,

-Health risk assessment, composition, and distribution of polycyclic aromatic hydrocarbons (PAHs) in drinking water of Southern Jharkhand, East India, Archives of environmental contamination and toxicology 80 (1), 120-133,

-Source identification and health risk assessment of atmospheric PM2. 5-bound polycyclic aromatic hydrocarbons in Jamshedpur, India, Sustainable cities and society 52, 101801

Conclusion: Authors should improve the work according comments. After above Minor revisions, I strongly recommended this paper for publication without any kind of hesitation in your journal.

The END

6. PLOS authors have the option to publish the peer review history of their article (what does this mean?). If published, this will include your full peer review and any attached files.

Reviewer #1: No

Reviewer #2: No

---

## [Author Response · Author response to Decision Letter 0]

2 Mar 2023

Reviewer 1 comments and Authors response

Although the analysis presented is sound, the description is not detailed enough. There needs to be a more thorough description of the results and their implications as well as a detailed background and context in the results section. Therefore, I recommend major revision before this manuscript can be accepted for publication in PLOS ONE.

Authors response: Thank you, for the comment we have made appropriate improvement in the results section. We have added the results of the clustering and outlier analysis with the data 

1. Results section: This section is too limited. It needs more details to add context to the results. For example, what does "significant spatial dependency" mean? What does it mean to say the PM concentrations are distributed non-random, clustered or dispersed. These results should be quantified and explained in more detail.

Authors response: We accepted the comment made necessary changes. Line number 275, 277-279, 283-286, 290-291, 293-299

2. Strengths and limitations section: Even though the satellite derived PM2.5 product has not been validated in the study region due to the lack of ground observations, it must have undergone validation by measurements in other locations. What are the conclusions of those validations. The authors need to discuss the general uncertainty in this product using literature review.

Authors response: We accepted the comment and discussed the results of these validations. Line number 363-369

3. The manuscript will improve with proof-reading and correcting grammatical and language errors. 

Authors response; We accepted the comment, gone through the manuscript and made changes

Line 28: This sentence is missing a verb. You might add "are" after "strong policies" 

Authors response: Thank you, we made the change. Line number 28

Line 30: It would be better to add "in Ethiopia" after "variation of PM2.5" in this sentence.

Authors response: Thank you, we made the change. Line number 30

Line 34: It is not clear why the “six years” is mentioned here. This is not consistent with the text which shows results for 22 years.

Authors response: Thank you, we made the change. Line number 35

Line 35: “Mann-Kendal” should be “Mann-Kendall”. Correct this spelling elsewhere too.

Authors response: Thank you, we made the change. Line number 36, 42, 54, 222, 224, 263

Line 44: Consider changing "has existed" to "exists".

Authors response: Thank you, we made the change. Line number 45

Line 48: Consider changing 'health-recommended' to 'WHO-recommended'.

Authors response: Thank you, we made the change. Line number 48

Line 49: Sentence fragment. Rewrite.

Authors response: Thank you, we made the change. Line number 50 

Line 94: “Outdoor PM2.5 level increases during winter, rainy, and spring seasons (3,4,18) than in summer and dry seasons” Is this a general statement or is it true for specific locations? Rainy seasons should see lower PM levels due to the scavenging of particulates from rain drops and cloud droplets. This should be clarified. 

Authors response: We accepted the comment and made explanations. Line number 91-105

Line 99 – Line 102: This statement requires citations. Here is a recent example with a focus on a South Asian country: https://doi.org/10.1016/j.atmosres.2021.105623

Authors response: Thank you, the statement is from the references cited in line 120. We made additional explanations. Line number 111- 114

Line 110: Subscript PM2.5 to be consistent with other places where it appears.

Authors response: Thank you, we made the necessary change. Line number 125, 52, 262

Line 166- Line 168: This statement should cite the ACAG satellite-based PM2.5 product publication.

Authors response: we accepted the comment and cited the appropriate reference. Line number 185

Line 268: Define what the high-high, low-low, high-low and low-high cluster and outliers mean. 

Authors response: Thank you, we have added explanations. Line number 290-291, 293-295, 296-297, 297-299

Reviewer 2 comments and authors response

1) The collection of lots number of Keywords, authors are requested to delete few and keep the genesis of this work.

Authors response: Thank you, we removed five keywords. Line number 53-54

(2) The numbering in the introduction should be avoided in the following lines: PM can be released straight into the air or formed from emitted pollutants (1). Chemicals constituted in PM include organic carbon, elemental carbon, soil dust, sulfate, nitrate, ammonium ions (2–4), ozone (5), and heavy metals (i.e. chromium and cadmium) (6). The biological constituents (bioaerosols) of PM2.5 include allergens and microbial compounds of bacterial and fungal origins (1,7).

Authors response: Thank you, we cited the references at the end of the paragraph. Line number 81

(3) Please change/modify the references in the following: “all-cause mortality (11,12,14–17)” 

Authors response: Thank you, we modified the references. Line number 90

4) Rephrase the following sentences It obligates the government to regulate, monitor, and oversee compliance. However, the government lacks enough monitoring laboratories and skilled professionals to monitor air quality. As a result, evidence that influences meaningful policy actions and record-keeping regarding air pollution is undersupplied (24). To the best of our knowledge, no study in the country characterizes the geographical distribution of ambient PM2.5. This study, therefore, has assessed the spatial and temporal variation of PM2.5 in Ethiopia.

Authors response: We accepted and rewrite the concept. Line number 134-143

(5) Abbreviations may be shifted into the starting of the manuscript.

Authors response: Thank you, we have gone through the manuscript and looked for abbreviations

(6) There are lots of commas and full stops are missing.

Authors response: Thank you, we have gone through the manuscript and made some changes

(7) Improve fig 1

Authors response: Thank you, we have improved the graph. Line Number 259, Fig1

(8) The current study may be reflect strong impact if you can cite the following reference. -Characterization of PM10 over urban and rural sites of Rajnandgaon, central India, Natural Hazards - Review on composition, emission sources of RSPM, TSPM, heavy metals and ions with effect on environment and health, Res. J. Chem. Environ -Understanding Sources and Composition of Black Carbon and PM2.5 in Urban Environments in East India - Toxicity and health risk assessment of polycyclic aromatic hydrocarbons in surface water, sediments and groundwater vulnerability in Damodar River Basin, Groundwater for Sustainable Development 13, 100553, -Health risk assessment, composition, and distribution of polycyclic aromatic hydrocarbons (PAHs) in drinking water of Southern Jharkhand, East India, Archives of environmental contamination and toxicology 80 (1), 120-133, -Source identification and health risk assessment of atmospheric PM2. 5-bound polycyclic aromatic hydrocarbons in Jamshedpur, India, Sustainable cities and society 52, 101801

Authors response: Thank you, we cited three of these references. Line number 91, 345

---

## [Decision Letter · Decision Letter 1]

9 Mar 2023

The spatial and temporal variation of fine particulate matter pollution in Ethiopia: Analysis of the Atmospheric Composition Analysis Group  data from 1998 to 2019

PONE-D-22-31123R1

Dear Dr. Shiferaw,

We’re pleased to inform you that your manuscript has been judged scientifically suitable for publication and will be formally accepted for publication once it meets all outstanding technical requirements.

Kind regards,

Basant Giri, Ph.D.

Academic Editor

PLOS ONE

Additional Editor Comments (optional):

Reviewers' comments:

Reviewer's Responses to Questions

**Comments to the Author**

1. If the authors have adequately addressed your comments raised in a previous round of review and you feel that this manuscript is now acceptable for publication, you may indicate that here to bypass the “Comments to the Author” section, enter your conflict of interest statement in the “Confidential to Editor” section, and submit your "Accept" recommendation.

Reviewer #1: (No Response)

Reviewer #2: All comments have been addressed

2. Is the manuscript technically sound, and do the data support the conclusions?

Reviewer #1: (No Response)

Reviewer #2: Yes

3. Has the statistical analysis been performed appropriately and rigorously? 

Reviewer #1: (No Response)

Reviewer #2: Yes

4. Have the authors made all data underlying the findings in their manuscript fully available?

Reviewer #1: (No Response)

Reviewer #2: Yes

5. Is the manuscript presented in an intelligible fashion and written in standard English?

Reviewer #1: (No Response)

Reviewer #2: Yes

6. Review Comments to the Author

Reviewer #1: (No Response)

Reviewer #2: Accepted in the current form. Author has replied satisfactory response of the reviewers comments. Recommended for publication in present form.

7. PLOS authors have the option to publish the peer review history of their article (what does this mean?). If published, this will include your full peer review and any attached files.

Reviewer #1: No

Reviewer #2: No

---

## [Editor Report · Acceptance letter]

16 Mar 2023

PONE-D-22-31123R1 

The spatial and temporal variation of fine particulate matter pollution in Ethiopia: Data from the Atmospheric Composition Analysis Group (1998-2019) 

Dear Dr. Shiferaw:

I'm pleased to inform you that your manuscript has been deemed suitable for publication in PLOS ONE. Congratulations! Your manuscript is now with our production department. 

Kind regards, 

on behalf of

Dr. Basant Giri 

Academic Editor

PLOS ONE